# Decoding critical long non-coding RNA in ovarian cancer epithelial-to-mesenchymal transition

Ramkrishna Mitra [1], Xi Chen[1], Evan J. Greenawalt[1], Ujjwal Maulik[2], Wei Jiang[3], Zhongming Zhao [4] & Christine M. Eischen[1]

Long non-coding RNA (lncRNA) are emerging as contributors to malignancies. Little is understood about the contribution of lncRNA to epithelial-to-mesenchymal transition (EMT), which correlates with metastasis. Ovarian cancer is usually diagnosed after metastasis. Here we report an integrated analysis of >700 ovarian cancer molecular profiles, including genomic data sets, from four patient cohorts identifying lncRNA *DNM3OS*, *MEG3*, and *MIAT* overexpression and their reproducible gene regulation in ovarian cancer EMT. Genome-wide mapping shows 73% of *MEG3*-regulated EMT-linked pathway genes contain *MEG3* binding sites. *DNM3OS* overexpression, but not *MEG3* or *MIAT*, significantly correlates to worse overall patient survival. *DNM3OS* knockdown results in altered EMT-linked genes/pathways, mesenchymal-to-epithelial transition, and reduced cell migration and invasion. Proteotranscriptomic characterization further supports the *DNM3OS* and ovarian cancer EMT connection. TWIST1 overexpression and *DNM3OS* amplification provides an explanation for increased *DNM3OS* levels. Therefore, our results elucidate lncRNA that regulate EMT and demonstrate *DNM3OS* specifically contributes to EMT in ovarian cancer.

[1] Department of Cancer Biology, Sidney Kimmel Cancer Center, Thomas Jefferson University, Philadelphia, PA 19107, USA. [2] Department of Computer Science and Engineering, Jadavpur University, Jadavpur 700032, India. [3] Department of Biomedical Engineering, Nanjing University of Aeronautics and Astronautics, Nanjing 211106, China. [4] Center for Precision Health, School of Biomedical Informatics, The University of Texas Health Science Center at Houston, Houston, TX 77030, USA. Correspondence and requests for materials should be addressed to C.M.E. (email: christine.eischen@jefferson.edu)

Ovarian cancer is the most lethal gynecologic malignancy in the United States with ~14,100 deaths and 22,500 new cases estimated for 2017[1]. The high mortality of ovarian cancer is primarily due to the high rate of therapy resistance and the diagnosis of the disease after it has metastasized, which occurs in ~80% of women[2,3]. Evidence indicates the ability of ovarian cancer cells to invade and metastasize is enhanced through the loss of epithelial features and the gain of a mesenchymal phenotype known as epithelial-to-mesenchymal transition (EMT)[4,5]. EMT leads to reversible reprogramming of cells, which is defined by fundamental changes initiated and maintained by critical genes, their regulatory circuits, and signaling pathways[6].

EMT is regulated at the transcriptional level by EMT-inducing transcription factors and at the post-transcriptional level by non-

coding RNA, such as microRNA (miRNA)[7]. Recently, reports suggest that long non-coding RNA (lncRNA), non-coding RNA longer than 200 nucleotides, can also modulate gene expression through several not fully characterized mechanisms[8]. LncRNA function in a broad range of cellular processes including cell proliferation, apoptosis, and reprogramming of cell pluripotency[9–11]. Aberrant expression of 15 lncRNA was detected in ovarian cancer tissue, but their functions were not investigated[12]. Elevated expression of one lncRNA, *ANRIL*, was reported to increase proliferation, migration, and invasion of serous ovarian cancer cells[13,14]. Additionally, 11 lncRNA have recently been shown to function in migration, EMT, and metastasis in multiple cancers; however, their role in ovarian cancer is unknown[15]. Therefore, much about lncRNA and ovarian cancer remains to be investigated.

In this study, we systematically identified lncRNA and their regulation of ovarian cancer EMT. Leveraging large-scale ovarian cancer molecular profiles and genomics from multiple independent patient cohorts revealed reproducible regulation of EMT by lncRNA in ovarian cancer. Verification, including evaluation of gene expression, levels of EMT proteins, and ovarian cancer cell migration and invasion after knockdown and proteotranscriptomic characterization, supported the link between EMT and lncRNA. This study increases understanding of lncRNA-mediated regulatory mechanism of ovarian cancer EMT and increases knowledge of the processes involved in ovarian cancer metastasis that may be targetable.

## Results

**Integrated analysis identifies EMT-associated lncRNA**. To determine whether lncRNA contribute to EMT in ovarian cancer, data from high-grade serous ovarian cancer patient samples were obtained from The Cancer Genome Atlas (TCGA) and stratified into 231 epithelial and 89 mesenchymal subtypes as previously defined[5] (patient subtype information in Supplementary Data 1). For this large cohort of patients, matched DNA methylation, gene copy number, and expression profiles of mRNA, lncRNA, and miRNA available in TCGA were utilized (Methods section; summarized in Fig. 1a, Supplementary Table 1). To predict EMT-linked lncRNA, we constructed a computational framework (Fig. 1). First we employed a multivariate linear regression model considering mRNA and lncRNA expression, copy number, and methylation profiling data from the matched patient samples of TCGA cohort and determined differentially expressed genes whose expression was significantly correlated with lncRNA expression (Benjamini–Hochberg (BH)[16] adjusted regression $P < 10^{-6}$; Fig. 1b, c). The regression model took into account the biases in estimating gene expression changes due to the corresponding copy number and DNA methylation changes (Methods section). In the spectrum of 386 protein coding genes that were significantly differentially expressed (twofold change; edgeR determined BH adjusted $P < 10^{-3}$) in the mesenchymal subtype

compared with the epithelial subtype, and 2959 non-differentially expressed genes (BH adjusted $P > 0.25$, as defined in ref. [17]) as background data, we inferred 25 lncRNA that had significantly enriched association (BH adjusted hypergeometric test $P < 10^{-3}$) with the differentially expressed genes (Fig. 1d). Of note, 44 differentially expressed genes known for inducing mesenchymal features including, EMT inducing transcription factors (*TWIST1*, *SNAI2*, *ZEB1*, and *ZEB2*), matrix metalloproteinases, BMP family genes, and collagen family genes (Supplementary Table 2), were identified indicating the regression model was built on meaningful data in the context of EMT[18–22].

We also identified the lncRNA that were differentially expressed in epithelial and mesenchymal lineage commitment and that are evolutionary conserved (Fig. 1e). We determined these 25 lncRNA had remarkably more significant differential expression (BH adjusted two-tailed *t*-test) in the mesenchymal subtype compared with the epithelial subtype than the remaining lncRNA (Supplementary Fig. 1a). Notably, seven lncRNA had at least twofold change in differential expression (*DNM3OS*, *MIAT*, *MEG3*, *DIO3OS*, *HAR1A*, *UCA1*, and *HCG14*; Supplementary Fig. 1b). The distribution of the expression of these seven lncRNA showed in both epithelial and mesenchymal subtypes that they were all well above the detection level that was previously determined for lncRNA expression[23] (Supplementary Fig. 2). Among the seven lncRNA, *UCA1* was not present in the 25 lncRNA list and lncRNA *HCG14* and *HAR1A* were poorly conserved across the primate species. The remaining four lncRNA (*DNM3OS*, *MIAT*, *MEG3*, and *DIO3OS*) had significantly elevated expression (BH adjusted two-tailed *t*-test) in the mesenchymal subtype and they were highly conserved (PhastCons conservation score = $0.94 \sim 0.98$) across the primate species (Fig. 1e, Supplementary Fig. 3a; Methods section). To determine whether the four lncRNA are present in normal ovarian cells, we extracted the expression of the four lncRNA in normal human tissues from the LncRNA2Function database[24]. All four lncRNA are expressed in normal ovary with *MEG3* having the highest and *MIAT* the lowest expression level (Supplementary Fig. 3b).

Differentially expressed genes, which were predicted to be regulated by one of the four lncRNA (*DNM3OS*, *MEG3*, *MIAT*, and *DIO3OS*) had significantly enriched association (BH adjusted hypergeometric test $P < 0.05$) with the EMT-linked canonical pathways, including focal adhesion, ECM-receptor interaction, and gap junction (Fig. 1f; Methods section). Because genes in the same pathway are typically co-expressed, we employed first-order partial correlation statistic to examine if the co-expression of the EMT-linked pathway genes was potentially induced by the inferred EMT-linked lncRNA[25] (Methods section). By definition, first-order partial correlation measures gene co-expression with removing the effect of a controlling variable; here the controlling variable is one lncRNA[26,27]. We observed a significant reduction of gene co-expression after removing the effect of mutually associated lncRNA ($P < 10^{-10}$, Wilcoxon rank-sum test; Supplementary Fig. 4), which indicate lncRNA-mediated regulation of

**Fig. 1** Identifying critical lncRNA in ovarian cancer EMT. **a** Ovarian cancer patients ($n = 320$) with genomic and molecular profiling data that classified into epithelial (Epi; $n = 231$) or mesenchymal (Mes; $n = 89$) subtypes were selected for analysis. **b** Heatmap of 386 genes that were differentially expressed in the mesenchymal subtype compared with the epithelial subtype. **c** Inferring deregulatory programs from ovarian cancer profiling data. Change in mRNA expression is modeled as linear function of the gene's DNA methylation, copy number, and lncRNA expression. **d**, **e** Systematic prediction of EMT-linked lncRNA from the lncRNA-gene association information obtained from the linear model. **d** The lncRNA that had significantly enriched association with the differentially expressed genes ($n = 25$, red dots; top 5 lncRNA labeled) were inferred as EMT associated. Remaining lncRNA were represented by gray dots. The X-axis with four different colors represent major annotation classes of the selected lncRNA ($n = 120$). The Y-axis denotes which lncRNA had enriched association with the differentially expressed genes compared with non-differentially expressed genes. **e** Filtering of high confidence EMT-linked lncRNA ($n = 4$; blue dots with labels) based on their aberrant expression (X and Y-axis) in EMT and conservation score (Z-axis). Gray dots represent remaining lncRNA. **f** Heatmap shows significantly enriched association of the inferred lncRNA with EMT-linked pathways. For **d** and **f**, P-values determined by BH adjusted hypergeometric test

**Table 1 Demographics and clinical information of ovarian cancer patient cohorts**

| Category (Number of samples) | TCGA[a,b,c] (320) | GSE9891[b,c,d] (233) | GSE18520[b,c] (53) | GSE26193[b,c] (100) | CPTAC[c] (103) |
|---|---|---|---|---|---|
| Subtype | | | | | |
| Epithelial | 231 | 136 | NA | NA | 71 |
| Mesenchymal | 89 | 97 | NA | NA | 32 |
| Histology | | | | | |
| Serous | 320 | 233 | 53 | 75 | 103 |
| Other | 0 | 0 | 0 | 25 | |
| Tumor grade | | | | | |
| I | 0 | 0 | 0 | 0 | 0 |
| II | 40 | 88 | All samples are high grade | 33 | 16 |
| III | 274 | 145 | All samples are high grade | 67 | 86 |
| IV | 1 | 0 | All samples are high grade | 0 | 0 |
| Undetermined | 5 | 0 | 0 | 0 | 1 |
| Tumor stage | | | | | |
| I | 0 | 10 | 0 | 17 | 0 |
| II | 18 | 9 | 0 | 9 | 7 |
| III | 252 | 193 | All samples are late stage | 58 | 78 |
| IV | 47 | 21 | All samples are late stage | 16 | 18 |
| Undetermined | 3 | 0 | 0 | 0 | 0 |
| Age at initial pathologic diagnosis | 30 ~ 87 | 23 ~ 80 | NA | NA | 34 ~ 87 |

[a]Discovery data
[b]Data used for survival analysis
[c]Data used for meta-analysis
[d]Independent validation data

EMT-linked pathway genes. Collectively, the data suggest the inferred lncRNA may have important roles in ovarian cancer EMT.

**Independent ovarian cancer data reproduce lncRNA regulation.** Reproducible regulation provides added confidence in the accuracy of the predictions and may reflect genuine molecular events[17,28]; therefore, we examined if the results obtained from TCGA data were consistent in another high-grade serous ovarian cancer patient cohort (Gene Expression Omnibus (GEO) accession ID: GSE9891; Table 1). This data set was stratified into 136 epithelial and 97 mesenchymal subtypes, as defined in Yang et al.[5] (Table 1, Supplementary Data 2). TCGA and this independent data set showed that a similar number of genes had significant differential expression (>twofold-change with BH adjusted $P <$ 0.05; 386 genes in TCGA and 346 genes in GSE9891 determined by edgeR and two-tailed $t$-test, respectively) in the mesenchymal subtype compared to the epithelial subtype. Additionally, the expression fold changes of the genes in these two data sets were strongly correlated (Spearman $\rho = 0.7$; correlation $P < 2.2 \times 10^{-16}$) in the spectrum of whole transcriptome data (Fig. 2a). Reannotation of microarray probe sets showed that $DIO3OS$, $DNM3OS$, $MIAT$, and $MEG3$ lncRNA were detected in the two ovarian cancer subtypes at levels similar to known protein coding EMT-linked genes (Supplementary Fig. 5). Except for $DIO3OS$, the other three lncRNA ($DNM3OS$, $MIAT$, and $MEG3$) had elevated expression in mesenchymal subtype compared to the epithelial subtype (Fig. 2b). These three lncRNA were strongly co-expressed (absolute Spearman $\rho > 0.3$; BH adjusted correlation $P < 10^{-4}$) preferentially with the genes that were differentially expressed in the two subtypes compared to the non-differentially expressed genes, but $DIO3OS$ did not (Fig. 2c). Subsequent pathway analysis revealed that $DNM3OS$, $MIAT$, and $MEG3$-associated differentially expressed genes were significantly enriched in the EMT-linked pathways (Fig. 2d; BH adjusted hypergeometric test $P < 0.05$). These data are consistent with the results obtained from TCGA.

To begin to evaluate the results obtained from our bioinformatics approach, we first focused on $MEG3$, which was reported to regulate EMT in lung cancer[29]. We examined genome-wide mapping of $MEG3$ binding sites, which were previously determined[30]. The data indicate that $MEG3$ potentially modulates the expression of 30 genes that are members of the EMT-linked pathways of which 22 genes had $MEG3$ binding sites at their proximal or distal regulatory regions. This is a ~2.6-fold enrichment (73.3% genes) compared with the total $MEG3$ bound genes in genome-wide scale (28.1%) (Fig. 3a, b; Methods section). Therefore, this $MEG3$ binding information verified the reliability of our prediction results and suggests direct regulation of EMT-linked genes by $MEG3$. Taken together, we observed highly reproducible lncRNA regulation in two independent patient cohorts, indicating the lncRNA $MEG3$, $DNM3OS$, and $MIAT$ likely have important roles in ovarian cancer cell EMT.

**$DNM3OS$ overexpression correlates with worse survival.** Given that overexpression of the three identified lncRNA potentially induces mesenchymal features, which contribute to metastasis, we questioned whether their overexpression would correlate with patient survival. To address this, we evaluated four independent ovarian cancer data sets (Table 1) and performed a 5-year survival analysis for each lncRNA separately. Patient samples were stratified based on the median expression of the specific lncRNA into high or low. There was no significant correlation of $MEG3$ or $MIAT$ overexpression with overall patient survival (Supplementary Fig. 6). However, three of the four patient cohorts showed that patients with higher $DNM3OS$ expression had significantly worse overall survival than those with lower $DNM3OS$ expression (Fig. 4; $P = 0.041$, $P = 0.033$, and $P = 0.054$, log-rank test for GSE9891, GSE18520, and GSE26193, respectively). There was a loss of 10, 17, and 16 months, respectively, in median survival for those patients with increased levels of $DNM3OS$ in the three data sets. Evaluation of genes associated with EMT showed that $E$-$CADHERIN$, $N$-$CADHERIN$, and $SNAIL$ expression were not correlated with overall survival, but $SLUG$ and $TWIST1$ expression were (Supplementary Fig. 7). We also assessed whether the expression of $DNM3OS$, $MEG3$, and $MIAT$ were linearly anti-correlated with survival time. $DNM3OS$, but not $MEG3$ and $MIAT$, had significant negative association (Spearman correlation

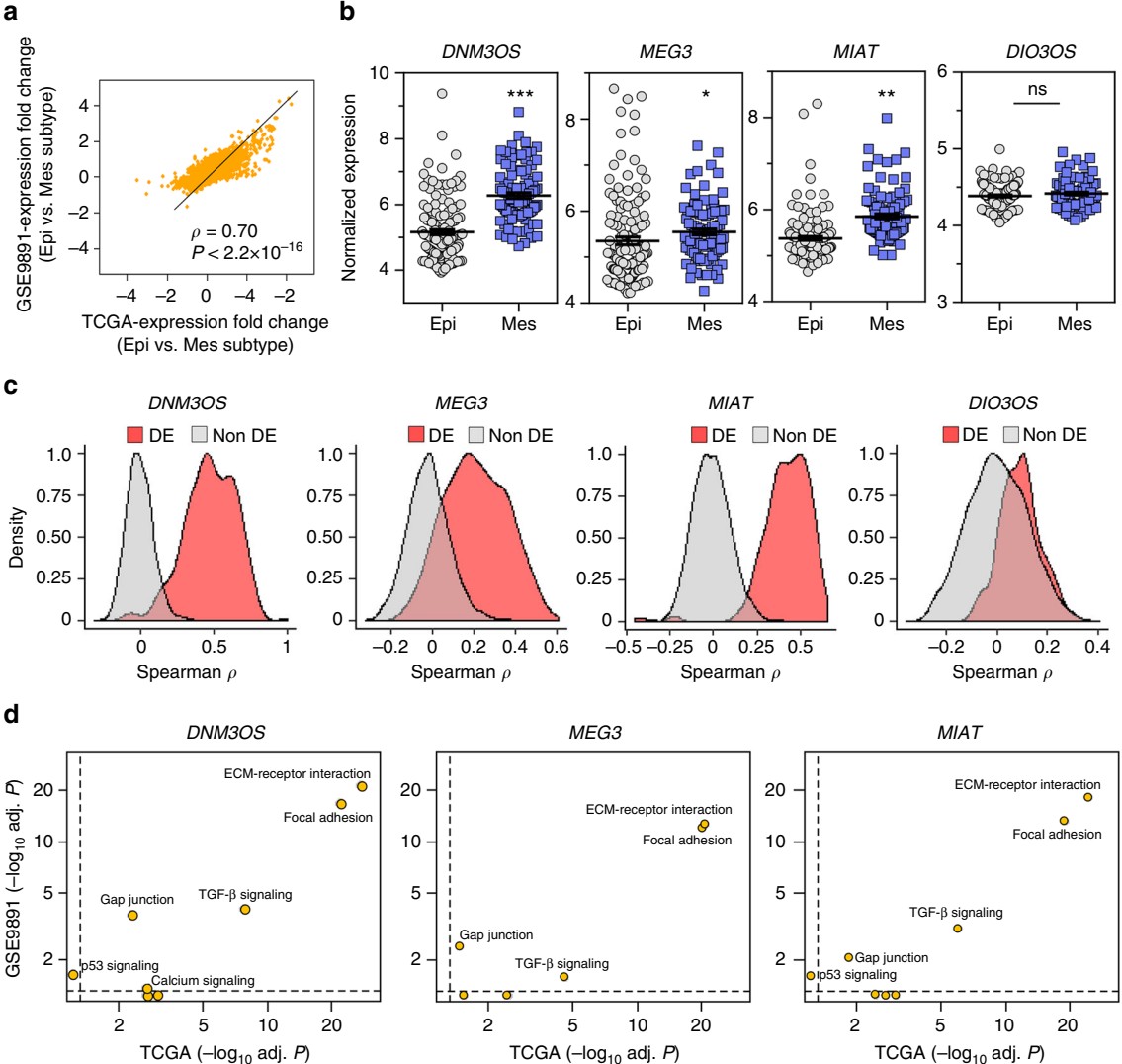

**Fig. 2** Reproducible lncRNA regulation in independent validation data. **a** Correlation of mRNA expression fold-changes between TCGA and GSE9891. In each dataset, fold changes were measured in the mesenchymal (Mes) subtype compared to the epithelial (Epi) subtype. **b** Expression differences of lncRNA in mesenchymal subtype ($n = 97$) compared to epithelial subtype ($n = 136$) in GSE9891. ***$P = 3.04 \times 10^{-18}$; **$P = 3.95 \times 10^{-11}$; *$P = 0.03$; NS = no significant difference, one-tailed $t$-test; mean ± SEM. **c** Expression correlation of the lncRNA indicated with differentially expressed (DE) or non-differentially expressed (non-DE) genes in GSE9891. **d** Pathway enrichment analysis for the differentially expressed genes that were co-expressed with lncRNA. *X*- and *Y*-axes represent enrichment *P*-value of EMT-linked pathway for TCGA and GSE9891, respectively. Dashed lines denote *P*-value threshold of 0.05 (i.e., 1.301 in −log₁₀ scale). *P*-values determined by BH adjusted hypergeometric test

$P < 0.05$) with survival time, and this was consistent for all the three data sets (Supplementary Table 3). Therefore, increased levels of *DNM3OS* correlates to poor overall survival of ovarian cancer patients, providing prognostic power of *DNM3OS*.

**Characterization of DNM3OS as a regulator of EMT genes.** We first selected *DNM3OS* associated 256 protein coding genes that were differentially expressed in the mesenchymal subtype compared with the epithelial subtype in TCGA cohort (Supplementary Table 4). We extracted physical relationships among these genes at their protein level from the curated human protein interaction database PINA v2.0[31]. Eighty-eight genes (34%) had at least one interaction with another *DNM3OS*-associated gene and in total, 132 interactions were determined (Supplementary Table 5). The networks had significantly enriched association (BH adjusted hypergeometric test $P < 0.05$) with several EMT-linked Gene Ontology biological processes and canonical signaling pathways (Supplementary Tables 6, 7). The results suggest the

differential gene expression in the EMT-linked protein interaction networks may be due to changes in *DNM3OS* expression. Second, we extracted *DNM3OS*-associated EMT-linked pathway genes identified in the TCGA cohort along with three additional EMT marker genes *E-CADHERIN* (*CDH1*), *N-CADHERIN* (*CDH2*), and *SNAIL* (*SNAI1*) and predicted their binding affinity with *DNM3OS*[32]. We observed that the distribution of minimum interaction energy between *DNM3OS* and the EMT-linked genes is significantly lower ($P = 7.43 \times 10^{-06}$; Kolmogorov–Smirnov test) compared with genome-wide *DNM3OS*-RNA interactions (Fig. 5a, Supplementary Fig. 8). To gain additional insight into *DNM3OS* regulation of EMT genes and determine whether *DNM3OS* has the potential to regulate the expression of EMT genes, we evaluated where *DNM3OS* resided in ovarian cancer cells. Cellular fractionation revealed *DNM3OS* is localized to the nucleus and not to the cytoplasm of ovarian cancer cells (Fig. 5b). Collectively, these results provide further support for *DNM3OS* regulating genes that mediate EMT.

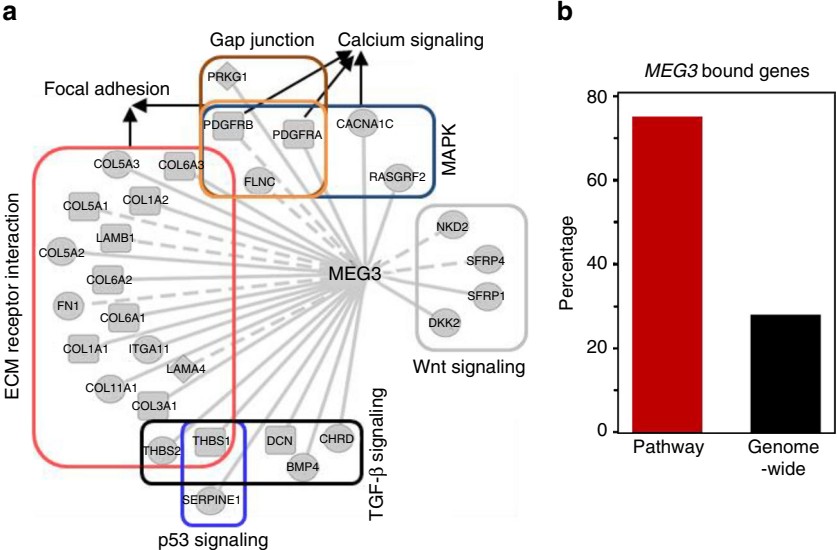

**Fig. 3** *MEG3* preferentially targets EMT-linked genes. **a** EMT-linked pathway genes having *MEG3* binding sites are represented by solid lines; remaining genes represented by dashed lines. Nodes with circle, diamond, and rectangle shapes represent predicted *MEG3* regulated genes as inferred from TCGA, GSE9891, or both data, respectively. **b** Enriched number of predicted *MEG3* regulated EMT-linked pathway genes had *MEG3* binding sites compared to the all known human genes

**Loss of DNM3OS induces mesenchymal-to-epithelial transition**. To further elucidate the contribution of *DNM3OS* in EMT in ovarian cancer and to experimentally validate our bioinformatics data, we evaluated knockdown of *DNM3OS* in ovarian cancer cells through multiple approaches. First, we performed whole transcriptome RNA-sequencing expression profiling after siRNA-mediated knockdown of *DNM3OS* in SKOV3 cells compared to non-targeting siRNA control (Fig. 6a). Gene set enrichment analysis (GSEA)[33] based on Kyoto Encyclopedia of Genes and Genome (KEGG) database[34] indicated *DNM3OS* knockdown results in deregulation of several EMT-linked pathways, such as regulation of actin cytoskeleton, focal adhesion, and WNT signaling pathways (Fig. 6b and Methods section). GSEA Hallmark data also showed deregulation of EMT process, Notch signaling and TGFβ signaling pathways in *DNM3OS* knockdown cells compared with the controls. Genes downregulated in *DNM3OS* knockdown cells (edgeR; at least twofold change with BH adjusted $P < 0.05$) were significantly enriched (BH adjusted hypergeometric test $P < 0.05$) with several EMT-linked pathways including focal adhesion, regulation of cytoskeleton, adherens, gap and tight junction, ECM-receptor interaction, and calcium and MAPK signaling pathways (Fig. 6c). These data indicate that these EMT pathways were preferentially deregulated with *DNM3OS* loss.

As a second approach, we performed western blot analysis of SKOV3 ovarian cancer cells after knockdown of *DNM3OS*. There were elevated protein levels of the epithelial marker E-CADHERIN, and reduced levels of the mesenchymal protein N-CADHERIN in the *DNM3OS* knockdown cells compared to control (Fig. 6d). Additionally, two transcription factors, SNAIL and SLUG that repress *E-CADHERIN* expression showed reduced levels in the *DNM3OS* knockdown cells (Fig. 6d). Therefore, our data indicate that loss of *DNM3OS* results in a mesenchymal-to-epithelial transition and alterations in these linked pathways.

Because mesenchymal cells typically have a greater capacity to metastasize[35], we next evaluated with transwell assays whether loss of *DNM3OS* would impact the ability of ovarian cancer cells to migrate and/or invade. *DNM3OS* knockdown in SKOV3 ovarian cancer cells resulted in significantly reduced numbers of cells migrating (Fig. 6e) and invading (Fig. 6f) as compared to cells with non-targeting control (two-tailed *t*-test). Because changes in proliferation rates can impact cell migration and invasion, we also assessed proliferation after *DNM3OS* knockdown. Rates of proliferation were analogous between the cells with *DNM3OS* knockdown and non-targeting control (Supplementary Fig. 9), indicating *DNM3OS* does not influence ovarian cancer cell growth. Therefore, *DNM3OS* regulates ovarian cancer cell movement, which is a critical contributor to metastasis.

We then performed proteotranscriptomic characterization of EMT-linked genes by conducting a meta-analysis to infer if the modulation of their expression related to the changes of *DNM3OS* expression. We assessed the siRNA-mediated *DNM3OS* knockdown data, three microarray gene expression profiling data sets available in the GEO database[36], RNA-sequencing data from TCGA, and protein expression profiling data from the Cancer Proteomic Tumor Analysis Consortium (CPTAC) (Table 1). For patient data, samples were stratified based on the highest or lowest quartile expression of *DNM3OS*. Gene expression changes were evaluated in patients that had the lowest quartile expression of *DNM3OS* compared with the patients that had the highest quartile expression of *DNM3OS*. We determined that genes involved in EMT were consistently deregulated in multiple data sets. Specifically, there was elevated expression of the epithelial marker gene *E-CADHERIN*, and decreased expression of mesenchymal marker genes *N-CADHERIN, SNAIL, SLUG, VERSICAN, VIMENTIN* in the samples with reduced *DNM3OS* (Fig. 6g). There were also reductions in the genes that induce or contribute to mesenchymal features such as collagen family genes, matrix metalloproteinase genes, and TGFβ pathway genes. Differential gene expression *P*-values, obtained from the multiple independent data sets, were summarized by computing the Fisher's combined probability test by using the R/Bioconductor package Survcomp[37]. The *P*-values represent overall significance of the differential expression of these important genes inferred from multiple independent data sets. Collectively, the results indicate that *DNM3OS* expression modulates ovarian cancer EMT by regulating the expression of several EMT-linked genes and their associated pathways.

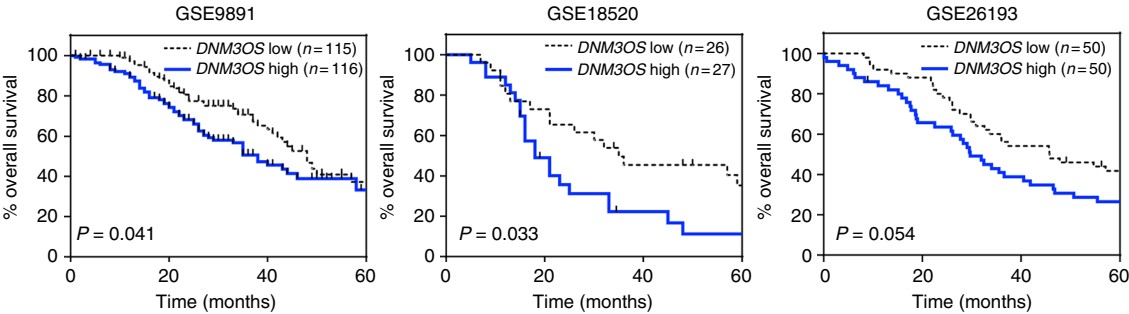

**Fig. 4** Elevated *DNM3OS* expression significantly correlates with poor overall survival of ovarian cancer patients. For individual data sets, ovarian cancer patient samples were separated into *DNM3OS* high and *DNM3OS* low based on the median expression of *DNM3OS*, and Kaplan–Meier survival analyses were performed. *P*-values determined by log-rank test

**Regulation of DNM3OS expression**. Among the 320 TCGA patients, 16 patients (5%) showed copy number amplification of *DNM3OS*, indicating that for these patients increased *DNM3OS* was presumably due to genomic amplification. However, the 304 patients that did not have *DNM3OS* amplification suggest that elevated *DNM3OS* expression in these patients is more likely to be the consequence of transcription factor dysregulation. The TWIST1 transcription factor was previously reported to regulate the *DNM3OS* locus. Specifically, in *Twist* knockout mice, the *DNM3OS* locus was downregulated[38]. Moreover, overexpression of TWIST1 in neuroblastoma cells resulted in increased *DNM3OS* levels, and luciferase reporter assays showed that the TWIST1-binding domain in the promoter of the *DNM3OS* locus was necessary for expression[39]. TWIST1 is overexpressed in ovarian cancer and a known ovarian cancer EMT marker[40,41]. In TCGA and independent validation (GSE9891) patient cohorts both *TWIST1* (Fig. 7a) and *DNM3OS* (Figs. 1e, 2b) were significantly overexpressed (>twofold change with BH adjusted *P* < 0.05; *TWIST1* in TCGA was determined by edgeR and remaining data were determined by two-tailed *t*-test) in the mesenchymal subtype compared with the epithelial subtype of ovarian cancer. We observed that knockdown of *TWIST1* resulted in significantly reduced levels of *DNM3OS* in SKOV3 ovarian cancer cells (Fig. 7b; two-tailed *t*-test *P* < 0.008). The shRNA that was the most effective in knocking down *TWIST1* resulted in the largest decrease in *DMN3OS*. Analysis of TCGA and an independent microarray mRNA expression profiling data set (GSE9891) both showed a strong positive correlation between *DNM3OS* and *TWIST1* (0.526 and 0.449, respectively; Spearman's *ρ*) (Fig. 7c). Our results together with previously published data indicate TWIST1 regulates the expression of *DNM3OS* in ovarian cancer cells.

In addition to transcriptionally inducing the expression of *DNM3OS*, TWIST1 also induces the miR-199/214 cluster, which resides within the human *DNM3OS* gene locus[32]. Analysis of miRNA-seq expression profiling data from the same TCGA patient cohort showed that the miRNA within the miR-199/-214 cluster (miR-214-5p, -214-3p, 199a-5p, and 199a-3p) were also significantly upregulated (>twofold change; BH adjusted *P* < 10$^{-18}$, determined by edgeR) in the mesenchymal subtype compared with the epithelial subtype (Fig. 7d). Analysis of TCGA matched miRNA-seq and mRNA-seq expression data revealed strong positive correlations for all four mature miRNA with *DNM3OS* (0.602 ~ 0.682; Spearman's *ρ*) and *TWIST1* (0.322 ~ 0.457; Spearman's *ρ*) (Fig. 7e). Therefore, overall, our results indicate lncRNA participate in ovarian cancer cell EMT, and specifically, increased *DNM3OS* expression by amplification or by TWIST1 overexpression contributes to EMT in ovarian cancer.

**Discussion**

Ovarian cancer is a deadly disease, and EMT is believed to be a significant contributor to its aggressiveness[42]. EMT is a complicated process that remains incompletely resolved, making it difficult to target therapeutically. Therefore, it is important to identify critical molecules that regulate EMT. In this study, we leveraged large-scale multidimensional TCGA genomic and protein expression data as well as multiple independent molecular profiling data for high-grade serous ovarian cancer to infer active lncRNA and their regulation potential in ovarian cancer EMT. Our comprehensive study identified three novel lncRNA (*DNM3OS, MEG3*, and *MIAT*) associated with ovarian cancer EMT. Genes predicted to be regulated by these lncRNA had significantly enriched association with the EMT-linked pathways. Several of these genes are known epithelial or mesenchymal markers whose reduced or elevated mRNA expression were strongly associated with expression changes of the inferred lncRNA in both TCGA and independent validation data. Additionally, genome-wide mapping of *MEG3* binding sites revealed that 73% of EMT-linked pathway genes that were deregulated in EMT in TCGA cohort are bound by *MEG3*, suggesting *MEG3* is likely involved in EMT in ovarian cancer. Previously, it was reported that *MEG3* regulated EMT in lung cancer[29]. *MIAT* had not been previously linked to EMT, but was shown to be upregulated in chronic lymphocytic leukemia and neuroendocrine prostate cancer[43,44]. Our experimental data showed alterations in *DNM3OS* expression were linked to EMT in ovarian cancer through changes in cell migration and invasion and EMT-linked RNA and protein levels, and ovarian cancer patient survival. Therefore, these specific lncRNA regulate EMT in ovarian cancer and likely contribute to metastasis and the high mortality of this disease.

One main issue in identifying EMT-linked lncRNA in large-scale data is to minimize false-positives. To achieve this goal, we started from the analysis of only 'known lncRNA' that are most reliable and well annotated in leading databases[45]. Second, we applied stringent thresholds to infer key lncRNA and their regulations. Finally, we required the lncRNA to be conserved across the primate species, which is an important filtering step since EMT is an evolutionary conserved process. More importantly, with the use of completely independent high-quality validation data, we highlighted lncRNA-mediated reproducible regulations in EMT. Reproducible results are expected to more likely reflect the true biological regulations in cellular system[17,28]. Due to the rapid growth of high-throughput genomic data, our integrated computational framework can be applied to other complex diseases for the purpose of deciphering their regulatory systems and identifying critical biomolecules.

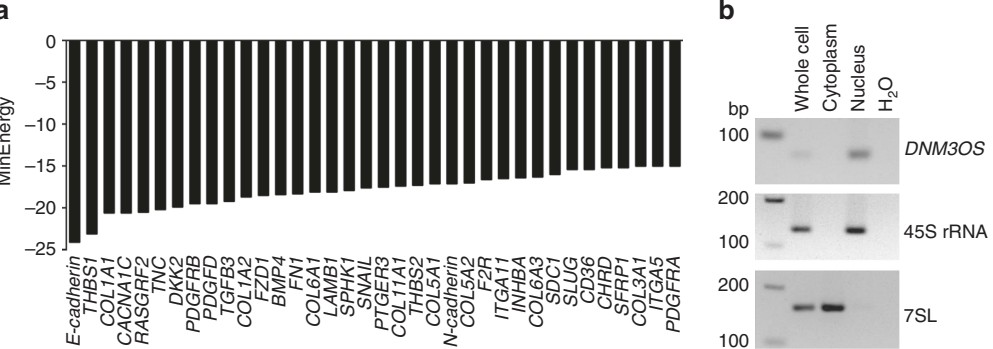

**Fig. 5** DNM3OS is a potential regulator of ovarian cancer EMT genes. **a** Interactions between EMT-linked genes and *DNM3OS* predicted by sequence complementarity and a minimum energy (MinEnergy) score ≤−15 kcal/mol. **b** Subcellular fractionation of RNA followed by RT-PCR (representative of two independent experiments). Nuclear 45S rRNA and cytoplasmic 7SL served as controls. Base pairs (bp) indicated on left side

*DNM3OS* was the top ranked deregulated lncRNA in ovarian cancer EMT, as well as the top ranked lncRNA among the lncRNA that had enriched association with the deregulated genes. *DNM3OS* has been reported to be a transcriptional target of TWIST1, which regulates ovarian cancer EMT[40,41], and was overexpressed in the mesenchymal subtype of ovarian cancer. Expression data also indicate a significant positive correlation between *DNM3OS* and *TWIST1* expression in ovarian cancer. We also showed that knockdown of *TWIST1* in ovarian cancer cells led to reduced *DNM3OS*. Additionally, the miR-199/-214 cluster encoded within the human *DNM3OS* gene locus[41] was also overexpressed in the mesenchymal subtype. Of note, elevated miR-214 expression promotes lung adenocarcinoma EMT, cell migration, invasion, and metastasis[46]. We showed that knockdown of *DNM3OS* reduces EMT-linked proteins and inhibited ovarian cancer cell migration and invasion, suggesting that miR-214 and *DNM3OS* may both contribute to EMT. Furthermore, increased expression of *DNM3OS* significantly correlated with reduced overall survival for patients with ovarian cancer. However, there was no correlation between patient survival and *MEG3* or *MIAT* levels. Therefore, of the three lncRNA associated with EMT we identified in our analyses, *DNM3OS* appeared to have a more significant impact. To determine the functional significance of altered *DNM3OS* levels in ovarian cancer cell EMT, we evaluated the effects on RNA and protein of knocking down *DNM3OS* in ovarian cancer cells. Pathway analysis of the differentially expressed genes from RNA-sequencing data revealed that multiple EMT-linked pathways were affected. Importantly, these pathways were highly overlapping with the pathways deregulated in the mesenchymal subtype compared with the epithelial subtype in both TCGA and an independent patient data set. Subsequent meta-analysis revealed that several known EMT markers had consistent expression changes that favorably induce EMT, in multiple independent data sets with the *DNM3OS* expression change. Moreover, Western blot results confirmed *DNM3OS*-mediated repression of epithelial markers and elevated expression of mesenchymal markers. Collectively, reproducible results both at the gene and pathway levels provide strong evidence for *DNM3OS* inducing ovarian cancer EMT. Taken together, our comprehensive analysis provides essential insights into ovarian cancer EMT and reveal the critical lncRNA and specifically, *DNM3OS* that regulate it. Our results open new avenues for targeting EMT in ovarian cancer.

## Methods
**Analysis of TCGA genomic and transcriptomic data**. Gene copy number and DNA methylation profiles, and lncRNA, mRNA, and miRNA expression profiles of high-grade serous ovarian cancer patient samples were extracted from TCGA. Normalized lncRNA and mRNA (RNA-seq) expression profiling data in terms of

reads per kilobase per million mapped reads (RPKM), and gene copy number data are available in Akrami et al.[47]. Additionally, we extracted and processed TCGA RNA-seq and miRNA-seq raw read counts to identify differentially expressed genes and miRNA, respectively. Level 4 TCGA DNA methylation data were extracted from the UCSC cancer browser[48]. Data from Illumina Infinium HumanMethylation27 platform were used since larger set of DNA methylation profiles was available for this platform compared to Illumina Infinium HumanMethylation450 platform at the time of analysis. Relative DNA methylation levels were measured as $\beta$-values ranging from 0 to 1 that represent the ratio of the intensity of the methylated bead type to the combined locus intensity. Here the $\beta$-values were offset by −0.5 to shift the whole data set to values between −0.5 and +0.5, as explained in ref. [48]. If multiple methylation probes mapped to a promoter region of a gene, a representative probe was selected that showed strongest negative correlation of methylation $\beta$-value and mRNA expression of that gene[49]. Genes commonly present in DNA methylation, gene copy number, and mRNA expression profiling data were selected for the analysis.

**TCGA ovarian cancer patient samples subtype classification**. A consensus clustering analysis of mRNA expression profiles of TCGA ovarian cancer patients classified four subtypes: differentiated, immunoreactive, proliferative, and mesenchymal[50]. Subsequently, considering miRNA-associated genes, Yang et al. reclassified the data into epithelial and mesenchymal subtype[5]. A fraction of the samples denoted as mesenchymal subtype by Yang et al. was classified as non-mesenchymal in original TCGA analysis, and these samples were excluded to retain high-quality data.

**Detection of lncRNA, mRNA, and miRNA in TCGA cohort**. To detect lncRNA, we applied two filters as mentioned in ref. [23]. First, lncRNA were eliminated if 50th percentile RPKM value = 0; second, lncRNA were selected if 90th percentile RPKM value is >0.1. Subsequently, 120 lncRNA with status 'known lncRNA' in GENCODE database (v21)[45] were selected for downstream analyses. We selected mRNA or miRNA if at least 75% of the samples had a normalized expression value ≥1.

**Differentially expressed mRNA, lncRNA, and miRNA in TCGA**. Both RNA-seq and miRNA-seq expression profiles of 231 epithelial and 89 mesenchymal patients were analyzed for identifying differentially expressed mRNAs and miRNA, respectively. We employed the R/Bioconductor package edgeR[51], which was designed to analyze digital gene expression data. Read counts were imported into edgeR for differential expression analysis. The data were normalized based on negative binomial distribution. Differential expression of mRNA or miRNA between mesenchymal and epithelial subtypes was assessed by estimating an exact test P-value, which is similar to the Fisher's exact test. The results were further adjusted using the Benjamini–Hochberg (BH) multiple testing correction method[16]. We quantified differential expression of lncRNA using t-test. Similar to mRNA or miRNA, P-values were adjusted using the BH method.

**Inferring lncRNA-associated mRNA in TCGA cohort**. The lncRNA-mRNA association strength was estimated by the corresponding coefficient score and P-values in a multivariate linear regression model. The model estimated mRNA expression changes due to copy number, DNA methylation, and lncRNA expression changes. Let $Y_{i,t}$ and $x_{k,t}^{lncRNA}$ denote the expression of gene $i$ and lncRNA $k$, respectively in sample $t$,

$$Y_{i,t} = \beta_0 + \beta_i^{DM} x_{i,t}^{DM} + \beta_i^{CN} x_{i,t}^{CN} + \beta_{i,k}^{lncRNA} x_{k,t}^{lncRNA},$$

where $\beta_0$ is the bias. $\beta_i^{DM}$ and $\beta_i^{CN}$ are the offsets accounting for the gene's expression changes at RNA level due to DNA methylation and gene copy number

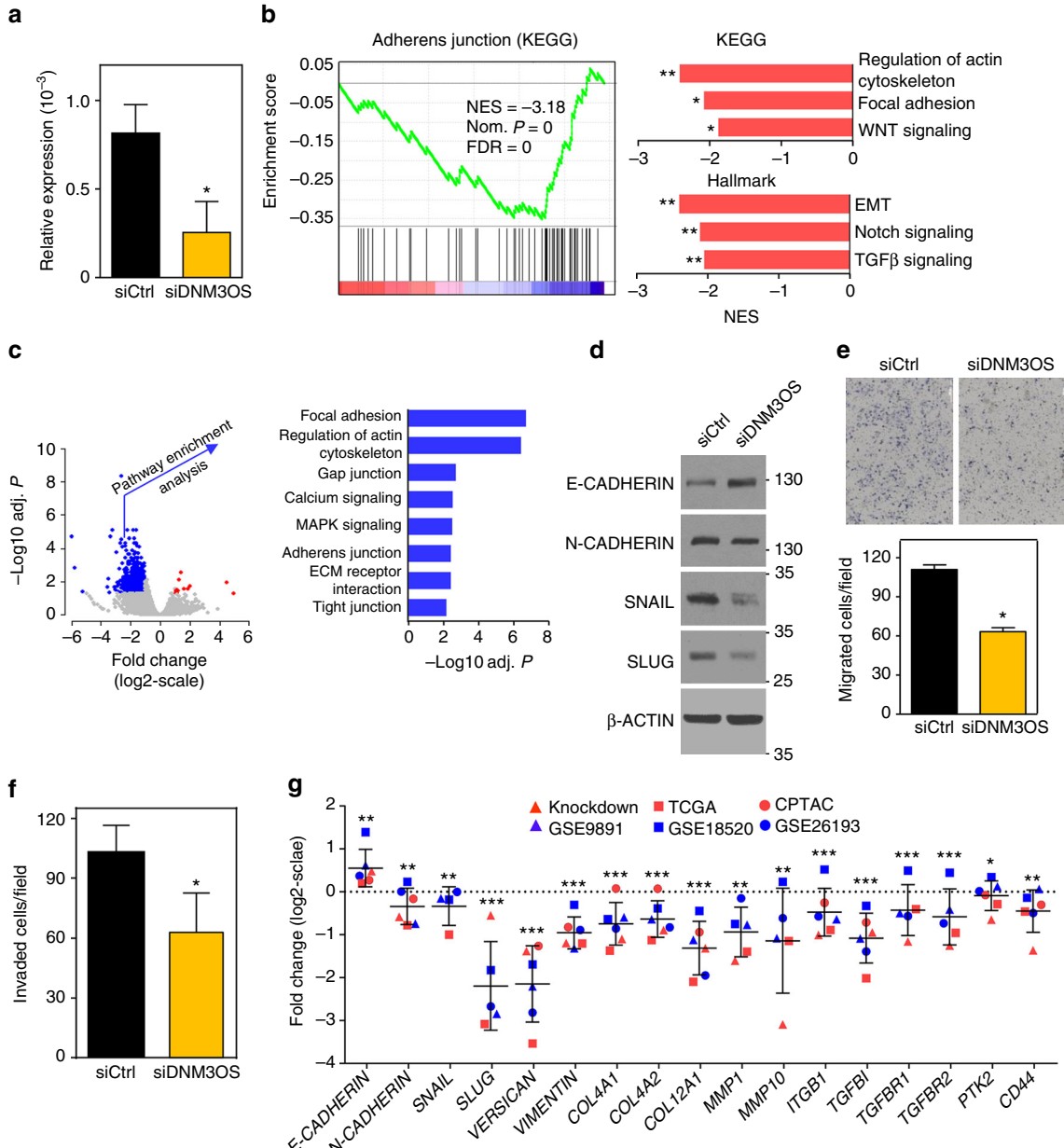

**Fig. 6** Reduced expression of *DNM3OS* favors mesenchymal-to-epithelial transition by deregulating critical genes/pathways. SMARTpool siRNA-specific *DNM3OS* or non-targeting (NT) control were transfected into SKOV3 cells. **a** qRT-PCR was performed. Values are relative to β-*ACTIN* and the mean of 3 experiments; *P = 0.039; one-tailed *t*-test. **b**, **c** RNA sequencing was performed (triplicates). GSEA on pooled samples considering curated data from KEGG and Hallmark data set **b**; normalized enrichment score (NES), nominal (Nom.) P-value, false discovery rate (FDR). **P < 0.006 and *P < 0.05; FDR. All nominal P < 0.006. Volcano plot (left) showing at least twofold up- (red dots) or downregulated (blue dots) genes in *DNM3OS* knockdown cells compared with non-targeting siRNA controls **c**. Gray dots represent genes that have lower than twofold expression change. Pathway enrichment analysis of significantly downregulated genes identified several EMT-associated pathways as deregulated in *DNM3OS* knockdown cells; P-values determined by BH adjusted hypergeometric test. **d**–**f** Western blot analysis **d**, transwell migration assay **e**; *P = 0.0002 (two tailed *t*-test), and transwell invasion assay **f**; *P = 0.0028 (two tailed *t*-test) were performed (all representative of at least three independent experiments). In **d**, representative pictures (4×) shown (white bar represents 300 μm). **g** Meta-analysis of six independent ovarian cancer data sets depicts expression changes of EMT-associated genes due to the reduced expression of *DNM3OS*; *P ≤ 0.01, **P ≤ 1.75 × 10^{-04}, ***P ≤ 1.43 × 10^{-12}; Fisher's combined probability test. For **a**, **e**, **f**, and **g** error bars are ± SEM

changes, respectively. The coefficient of interest $\beta_{i,k}^{lncRNA}$ indicates the association strength between lncRNA *k* and gene *i*. Regression P-values were adjusted using multiple test correction method (FDR) and significantly associated lncRNA and mRNA pairs were selected after applying the strict criteria of regression coefficient cutoff ±0.3 and BH adjusted P < 10^{-6}.

**Inferring the impact of lncRNA in gene co-expression**. Let us assume that, at the transcript level, a co-expressed gene pair has two genes, $g_x$ and $g_y$. Let us also assume that one lncRNA is mutually co-expressed with $g_x$ and $g_y$. The Spearman's $\rho$ is denoted by $r_{g_xg_y}$. The first-order partial correlation between $g_x$ and $g_y$

conditioning on lncRNA is:

$$r_{g_xg_y.lncRNA} = \frac{r_{g_xg_y} - r_{g_xlncRNA}r_{g_ylncRNA}}{\sqrt{(1 - r_{g_xlncRNA}^2)(1 - r_{g_ylncRNA}^2)}}.$$

**Genome-wide *MEG3* binding sites inferred by ChOP-Seq**. We obtained genome-wide mapping of *MEG3* binding sites determined by a modified chromatin oligo affinity precipitation (ChOP) method from the author[30]. As explained in the original study, we identified genes having *MEG3* bound genomic regions using the tool GREAT[52].

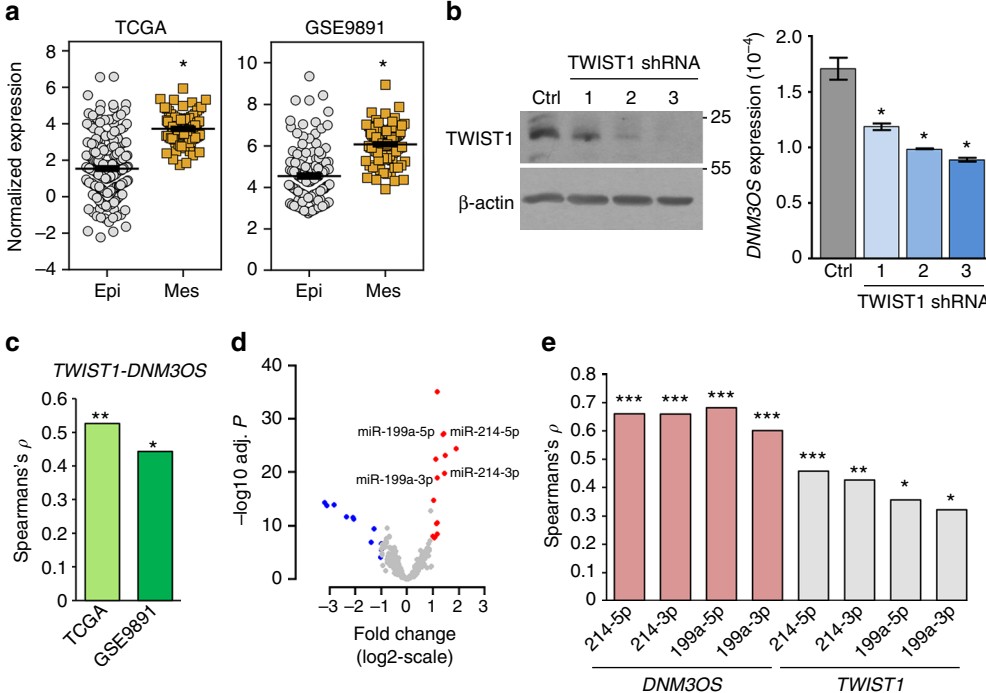

**Fig. 7** *TWIST1 regulates DNM3OS expression in ovarian cancer cells.* **a** Expression of *TWIST1* in mesenchymal (Mes; $n = 89$ for TCGA and $n = 97$ for GSE9891) subtype compared with the epithelial (Epi; $n = 231$ for TCGA and $n = 136$ for GSE9891) subtype in two independent patient cohorts. BH adjusted *$P < 10^{-19}$; determined by edgeR for TCGA and two-tailed *t*-test for GSE9891; mean ± SEM. **b** SKOV3 ovarian cancer cells expressing one of three *TWIST1* shRNA or control non-targeting shRNA were Western blotted and *DNM3OS* levels were evaluated by qRT-PCR (data presented are one of two independent experiments performed); error bars are ± SEM *$P < 0.008$; two tailed *t*-test. **c** Spearman's correlation coefficient (y-axis) between *TWIST1* and *DNM3OS* expression in two data sets (x-axis); *$P = 4.16 \times 10^{-13}$, **$P < 2.2 \times 10^{-16}$. **d** Volcano plot of miRNA in TCGA mesenchymal subtype compared with epithelial subtype that are at least twofold upregulated (red dots) or downregulated (blue dots). Gray dots represent remaining miRNA. miRNA in the miR-199/-214 cluster are labeled. *X*-axis is miRNA expression fold-change, and *Y*-axis is differential expression *P*-values determined by edgeR. **e** Expression correlation between *DNM3OS* and miRNA or *TWIST1* and miRNA in TCGA cohort. *X*-axis represents *DNM3OS* or *TWIST1* and indicated miRNA pair. *Y*-axis represents Spearman's rank correlation coefficient (Spearman's $\rho$) for each pair. *$P < 3.6 \times 10^{-09}$; **$P = 1.41 \times 10^{-15}$; ***$P < 2.2 \times 10^{-16}$

**Detection of conserved lncRNA.** A sliding window of 200 nucleotides was chosen along the lncRNA transcript and conservation levels were measured using the average phastCons score across the primate species. The maximally conserved 200 nucleotides sliding window was selected as the representative conservation score[53].

**Pathway enrichment analysis.** Differentially expressed genes significantly co-expressed with a specific lncRNA were selected to conduct pathway enrichment analysis using the canonical KEGG pathway database[34], which is embedded into the software WebGestalt[54]. Pathways showing significantly enriched (BH adjusted hypergeometric test $P < 0.05$) number of differentially expressed genes were recognized as the lncRNA associated pathways.

**Analysis of microarray mRNA and lncRNA expression data.** Three independent microarray gene expression profiling data (accession numbers GSE9891, GSE18520, and GSE26193) for ovarian cancer patients were downloaded from the GEO database[36]. The data were generated on the Affymetrix Human Genome U133 Plus 2.0 platform following standard Affymetrix protocol. The data were normalized using the robust multi-array average (RMA) algorithm[55] in the Affy package for R[56]. We mapped the probe set IDs to the NetAffx annotation file to extract lncRNA expression. Probes were averaged to calculate a single-expression intensity measure per gene/lncRNA per array. Differential expression was measured using unpaired *t*-tests. lncRNA and mRNA association was established by Spearman rank correlation.

**siRNA, shRNA, and western blotting.** Human ovarian cancer SKOV3 cell lines (mycoplasma free, authenticated) from Dr. Dineo Khabele were cultured as previously described[57] in RPMI1640 media, 10% fetal bovine serum, penicillin, and streptomycin. SKOV3 cells were transfected with 50 nM of non-targeting control or *DNM3OS* siRNA Smartpool (Dharmacon) containing 4 siRNA using Lipofectamine 2000 (Thermo Fisher Scientific). Lysate from SKOV3 cells grown in ≤2% fetal bovine serum was prepared as previously reported[59] with RIPA-buffer containing protease inhibitors. For *TWIST1* knockdown, SKOV3 cells were infected with *TWIST1*-specific shRNA encoded lentivirus or non-targeting control[59] (obtained from Dr. Andrew Aplin, Thomas Jefferson University); 3 days after

puromycin selection, cells were lysed as indicated above. Proteins were subjected to SDS-PAGE, transferred to nitrocellulose, and Western blotted with the following antibodies: E-CADHERIN, N-CADHERIN, SNAIL, and SLUG (1:1000; Cell Signaling), TWIST1 (1:500; Santa Cruz), and β-ACTIN (1:2000; Sigma). Uncropped scans of the blots are in Supplementary Fig. 10.

**Cell migration and invasion analysis.** SKOV3 ovarian cancer cells were transfected with *DNM3OS* siRNA or non-targeting control (described above), re-suspended in serum free RPMI1640. For migration assays, cells were placed into upper chamber inserts (70,000 cells per well, 8.0 μm pores, BioCoat insert). For invasion assays, cells were seeded in matrigel invasion chambers (150,000 cells per well, 8.0 μm pores, Corning BioCoat Matrigel invasion chamber). After 18 h (migration) or 24 h (invasion) incubation at 37 °C, migrated or invaded cells, respectively, were fixed and stained with Siemens Diff-Quik stain kit (Siemens Healthcare Diagnostic). Using a fixed grid, migrated and invaded cells were counted in 4 and 8 independent fields, respectively, using an Olympus CKX53 inverted microscope (×10), and images were taken by Cytation 5 cell imaging reader (×4 magnification).

**qRT-PCR and RNA sequencing.** Total RNA was isolated with Trizol (Sigma) from SKOV3 cells transfected with *DNM3OS* siRNA or non-targeting control and cDNA generated as we previously reported[60] with Superscript III (Invitrogen) as per manufacturer's instructions. qRT-PCR (triplicates) performed for *DNM3OS* and β-*ACTIN* as previously described[60] with RT² SYBR Green qPCR Mastermixes (Qiagen). *DNM3OS* forward: 5′-GGTCCTAAATTCATTGCCAGTTC-3′ and reverse: 5′-ACTCAAGGGCTGTGATTTCC-3′ primers. RNA sequencing profiles (48 h; triplicates) were generated using the Illumina TrueSeq platform. Maximum likelihood estimates of transcript read count for each sample were computed with Kallisto v0.43.0[61]. Human transcriptome fasta file was downloaded from Kallisto (Ensembl version GRCh38) and used to create a Kallisto index. On paired fastq files, for each sample, the Kallisto quantification algorithm was run to estimate transcript abundance. Using tximport the transcript level abundances were summarized into gene level abundances[62]. The read counts obtained from tximport were imported into edgeR for differential expression analysis[51]. In edgeR, data were

normalized based on negative binomial distribution. The differential expression of genes between *DNM3OS* knockdown and non-targeting control samples was assessed by estimating an exact test *P*-value.

**lncRNA cellular localization evaluation.** The nuclear and cytoplasmic RNA from SKOV3 cells were separated as previously reported[63], except lysis buffer was 10 mM Tris (pH 8.0), 140 mM NaCl, 1.5 mM MgCl₂, 0.5% Igepal, 3 μl/ml RNa-seout. cDNA was synthesized and RT-PCR performed as described above. Primers included the *DNM3OS* primers described above, 7SL primers previously reported[64], and 45S rRNA forward: 5′-GTCAGGCGTTCTCGTCTC-3′ and reverse 5′-CACCACATCGATCACGAAGA-3′ primers.

**GSEA.** We conducted GSEA[33] using the KEGG suite[34] and the Hallmark gene set embedded in GSEA to identify signaling pathways that were differentially activated in *DNM3OS* knockdown cells compared to non-targeting siRNA control cells. GSEA was run on pre-ranked list of the genes obtained from the *edgeR* analyses. Ranking was based on fold-change induction in *DNM3OS* knockdown cells compared to control cells. GSEA results were assessed as being statistically significant by permutation of 1000 samples.

**Data availability.** The RNA-sequencing data have been deposited in the Gene Expression Omnibus (GEO) and NCBI Sequence Read Archive (SRA) databases under the accession code GSE104295 and SRP118934, respectively. All other publicly available data referenced during the study can be retrieved from GEO (https://www.ncbi.nlm.nih.gov/gds/), TCGA (https://cancergenome.nih.gov/), or respective authors' websites. The other data supporting the findings of this study are available within the article and its Supplementary Information and Supplementary Data files.

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

## Acknowledgments

We thank the MetaOmics facility supported by the NCI Cancer Center support grant P30CA056036 for RNA-seq, Drs. Andrew Aplin and Edward Hartsough for *TWIST1* shRNA, Dr. Mick Edmonds for advice on cell migration and invasion analysis, and members of the Eischen lab for helpful suggestions. Support for this study was provided by NCI R01CA177786 (C.M.E.), the Pellini Foundation Fund, and the Sidney Kimmel Cancer Center.

## Author contributions

R.M. and C.M.E. designed the study; R.M. performed the bioinformatic analyses; W.J. performed initial TCGA RNA-seq analyses; U.M. performed methylation analysis; X.C. performed the knockdown experiments; E.G. performed the transwell assays; Z.Z. supervised the initial TCGA RNA-seq analysis; R.M. and C.M.E. wrote the manuscript; and all authors read and approved the manuscript.

## Additional information

**Competing interests:** The authors declare no competing financial interests.

