## [Peer Review File · Nature Communications]

Reviewers' comments:

Reviewer #1 Expert in regulatory RNAs and cancer:

This study investigated a potential role of the lnc DNM3OS in regulation of EMT in ovarian cancer. The authors provide evidence from multiple cohorts that DNM3OS is overexpressed in ovarian cancer. Although the topic is interesting, majority of the authors findings are highly correlative. The authors did not characterize this lnc in the ovarian cancer cells. This is important given the fact that lncs are often expressed and sometimes alternatively in a cell-type specific manner. Is DNM3OS nuclear or cytoplasmic lncRNA? What is its copy number per cell? It is unclear if the observed changes in expression of EMT markers upon knockdown of DNM3OS are due to direct effect of this lnc or due to indirect or nonspecific (only one siRNA was used) effects. Moreover, no evidence was provided on the phenotypic changes such as cell migration, invasion or induction of MET after knockdown of this lnc. Finally, the association between TWIST and DNM3OS is also correlative and no evidence is provided comparing the effects of TWIST and DNM3OS in the cells used in this study. In my opinion, this work may be more suitable for another journal.

Reviewer #2 Expert in cancer epigenomics:

This is an interesting paper in characterizing the function of lncRNAs in EMT and their clinical implications. Below are some issues to be addressed for its clarification and soundness.

- 1) The authors need to provide more detail for the lncRNA data used in analysis: how were they detected and how many of them were included based on which gene annotation?
- 2) In inferring lncRNA-associated mRNA in TCGA cohort, how lncRNA was selected to be included in the model for a particular mRNA gene? I doubt all combinations were analyzed.
- 3) For the most interested lncRNAs identified (for example the seven with fold change > 2), what are their expression level like in two subtypes, i.e., their percentile of expression in each type. This is very informative as many lncRNAs are expressed at very low level and their clinical relevance can be less significant if too low.
- 4) MEG3 and MIAT are associated with EMT but they are not associated with survival. What is the explanation or speculation from the authors? What is the correlation between MEG3 and DNM3OS?
- 5) Did the authors try the linear expression association with survival time for the three lncRNAs (not by median based binary)?
- 6) Also, in the survival association analysis, only univariate was performed. The authors should conduct multivariable analysis with inclusion of other variables (for example stage, treatment, subtype and tumor grade).
- 7) What are protein coding genes significantly associated with DNM3OS expression? What is their physical relationship between? Any adjacent target?
- 8) If DNM3OS is the target of TWIST1 and it is also coordinated with miR-214, the authors should provide their correlation information.

We thank the reviewers for their time and critiques. We have addressed each reviewer comment point-by-point below and have altered the text accordingly. We have added 18 new pieces of data in response to the reviewer comments (Figures 5A, 5B, 6E, 6F, 7B, 7C, 7E; Supplemental Figures 2, 3B, 5, 7, 8, 9; Supplemental Tables 5, 6, 7, 8, and 9).

Reviewer 1:

The authors did not characterize this lnc in the ovarian cancer cells. This is important given the fact that lncs are often expressed and sometimes alternatively in a cell-type specific manner.

We have performed multiple additional experiments to further characterize *DNM3OS* in ovarian cancer cells. Our experiments evaluated *DNM3OS* localization, function, and regulation (see responses below). The results provide additional validation of *DNM3OS* in EMT and significantly increased understanding of *DNM3OS* in ovarian cancer cells.

Is DNM3OS nuclear or cytoplasmic lncRNA?

We have evaluated *DNM3OS* localization in ovarian cancer cells. Using a cell fractionation approach, our data show that *DNM3OS* is localized to the nucleus and is not in the cytosol of SKOV3 ovarian cancer cells. These data are now Figure 5B. Additionally, an internet search showed an abstract from a recent FASEB meeting that indicates *DNM3OS* was localized to the nucleus in macrophages (Das et al *April 2017 FASEB Journal 31:1 Supplement 757.6*), further supporting our data of nuclear localization of *DNM3OS*.

What is its copy number per cell?

We are unclear about this question, as copy number is not directly linked to function or importance. Lowly abundant molecules can be equally important to a cell as abundant molecules, particularly if they are the rate limiting factor. However, we do recognize that lncRNA can be expressed at very low levels, which could possibly indicate that it may be less clinically relevant. Therefore, to determine the level of *DNM3OS* expression, we have taken multiple approaches. Firstly, we have determined experimentally that *DNM3OS* is expressed at detectable levels in ovarian carcinoma cells without additional filtering or concentration techniques (Figure 6A), suggesting that it is not expressed at very low levels. Secondly, by analyzing lncRNA expression profiles from TCGA and GSE9891, we determined that *DNM3OS* is expressed well above baseline levels in ovarian cancer cells with the mesenchymal subtype expressing significantly more than the epithelial subtype (Supplemental Figures 2 and 5). Also, please see our answer to reviewer #2 comment 3. Finally, we determined that *DNM3OS* is expressed in normal ovary and its levels are higher in normal ovary than other normal tissues (Supplemental Figure 3B). Together, our data indicate that *DNM3OS* is expressed in the ovary normally and that there is increased expression in the mesenchymal compared to the epithelial subtype of ovarian cancer cells. We have added the new data to the manuscript.

It is unclear if the observed changes in expression of EMT markers upon knockdown of DNM3OS are due to direct effect of this lnc or due to indirect or nonspecific (only one siRNA was used) effects.

We apologize that we did not clearly indicate that the siRNA we used was a pool of 4 siRNA that target different regions of *DNM3OS*. We have corrected the Methods section to denote this. Additionally, *DNM3OS* is approximately 8 kilobases in size, making experiments that require its ectopic overexpression difficult to do. Therefore, we have performed an analysis to identify genes that *DNM3OS* has the potential to bind due to sequence complementary and interaction energy. We observed that the distribution of minimum interaction energy between *DNM3OS* and multiple EMT-linked genes is significantly lower ($P=7.43 \times 10^{-06}$; Kolmogorov-Smirnov test) compared with genome-wide *DNM3OS*-RNA interactions. These data are now Figure 5A and Supplemental Figure 8. These data together with the data that an established regulator of EMT, TWIST1, regulates *DNM3OS* expression (see below), that *DNM3OS* resides in the nucleus (see above), that *DNM3OS* contributes to cell migration and invasion (see below), and the analysis of independent data sets indicate *DNM3OS* is linked to EMT gene expression provide multiple lines of evidence that *DNM3OS* regulates the expression of one or more EMT-linked genes.

Moreover, no evidence was provided on the phenotypic changes such as cell migration, invasion or induction of MET after knockdown of this lnc.

We have performed additional experiments to evaluate phenotypic changes in ovarian cancer cells following knockdown of *DNM3OS*. Specifically, we have performed both migration and invasion assays. Our data show that loss of *DNM3OS* results in a significant reduction in the number of ovarian cancer cells that migrated and invaded as compared to cells that received the non-targeting control. These data have been added to the manuscript and are now Figure 6E and 6F. Because changes in proliferation could impact migration and invasion, we also evaluated growth rates of ovarian cancer cells following *DNM3OS* knockdown. Our data show rates of proliferation of SKOV3 ovarian cancer cells between *DNM3OS* knockdown and non-targeting control were analogous. These data are now in supplementary Figure 9. These new results show that loss of *DNM3OS* negatively impacts cell migration and invasion, but not proliferation, and support our conclusions that *DNM3OS* contributes to EMT.

Finally, the association between TWIST and DNM3OS is also correlative and no evidence is provided comparing the effects of TWIST and DNM3OS in the cells used in this study.

We apologize that we did not sufficiently explain previous studies that indicate the *DNM3OS* locus is regulated by TWIST1 and that TWIST1 is an established regulator of ovarian cancer EMT. The *DNM3OS* locus was identified as being regulated by Twist while evaluating *Twist* knockout mice (Loebel et al *Genesis* 2002). Moreover, overexpression of TWIST1 in neuroblastoma cells resulted in increased *DNM3OS* levels, and luciferase reporter assays showed that the TWIST1-binding domain in the promoter of the *DNM3OS* locus was necessary for expression (Lee et al *Nuc Acids Res* 2009). We have included a description of these previous studies in the Results section on page 12 to better put in context our experiments and results. Additionally, with many publications on ovarian carcinoma, it has been established that TWIST1 regulates EMT, is overexpressed in histological samples, and contributes to metastasis and poor survival for patients with this malignancy (e.g., Cho et al. *J Immunol* 2016, Kim et al *Korean J Path* 2014, Zhu et al. *Oncogene* 2014, Hosono et al. *Br J Cancer* 2007, Nuti et al. *Oncotarget* 2014, Ren et al. *J Cancer* 2016, Wushou et al. *Int J Mol Sci* 2014, Zhou et al. *Tumor Biol* 2014, etc.). Because of these previous publications and that the SKOV3 cells we used in our study were

used in several of the previous publications, we did not think there was sufficient scientific rationale to extensively experimentally test the association between *TWIST1* and *DNM3OS*. However, we have now also evaluated the effects of *TWIST1* knockdown in our SKOV3 ovarian cancer cells after obtaining previously published *TWIST1* shRNA (Weiss et al. *Cancer Res* 2012). We observed that knockdown of *TWIST1* in SKOV3 ovarian cancer cells resulted in a significant reduction in *DNM3OS* (see Figure 7B). Reduced levels of *DNM3OS* resulted in increased levels of E-CADHERIN, decreased levels of N-CADHERIN, SNAIL, and SLUG (Figure 6D), and reduced ovarian cancer cell migration and invasion (Figure 6E and 6F). Additionally, we have added data analyses showing strong positive expression correlations between *TWIST1* and *DNM3OS* in two independent ovarian cancer datasets (Figure 7C). We have also evaluated *TWIST1* expression in ovarian cancer patient overall survival. These data show that analogous to increased *DNM3OS*, increased *TWIST1* expression correlates to significantly reduced overall survival (Supplementary Figure 7). These results combined with the other results in the manuscript and the results from other groups described above support our conclusions that *TWIST1* is a transcription factor that regulates *DNM3OS* expression and *DNM3OS* regulates EMT.

Reviewer 2:

1) The authors need to provide more detail for the lncRNA data used in analysis: how were they detected and how many of them were included based on which gene annotation?

In the revised version in the Method section on page 17, we added a subsection ‘Detection of lncRNA, mRNA, and miRNA in TCGA cohort’ to provide more detail for the lncRNA data used in our analyses. We indicated how the lncRNA were detected and how many of them were included based on GENCODE (v21) gene annotation database.

2) In inferring lncRNA-associated mRNA in TCGA cohort, how lncRNA was selected to be included in the model for a particular mRNA gene? I doubt all combinations were analyzed.

Our computational framework started with building a multivariate regression model that integrated the expression of protein coding genes and only the known lncRNA (to minimize false positive predictions), DNA copy number and methylation profiles from a large-scale matched patient samples from TCGA. Thereafter, we employed a combination of different stringent filtering steps for prioritizing the EMT linked lncRNA in ovarian cancer. We filtered the most promising lncRNA based on their 1) significantly enriched association with the dysregulated protein coding genes in EMT, 2) differential expression in mesenchymal subtype compared with the epithelial subtype, and 3) conservation score across the primate species, which is relevant in the context of evolutionary conserved process EMT. Of note, among the significantly dysregulated protein coding genes included in the model system, 44 genes are known for inducing mesenchymal features (enlisted in Supplementary Table 3), indicating the regression model was built on meaningful data in the context of EMT. The entire computational framework was designed very carefully with stringent criteria to reduce the false positive predictions. We describe the framework on page 4 of the Results under the subsection ‘Integrative computational framework identifies EMT-linked lncRNA in ovarian cancer’ and in the Methods section on page 16.

3) For the most interested lncRNAs identified (for example the seven with fold change > 2), what are their expression level like in two subtypes, i.e., their percentile of expression in each type. This is very informative as many lncRNAs are expressed at very low level and their clinical relevance can be less significant if too low.

In the revised version of the manuscript, we added empirical cumulative distribution of the expression of 7 lncRNA in the mesenchymal and epithelial ovarian cancer subtypes. The results show that the selected lncRNA had considerably higher expression level compared with the lncRNA detection level (Yan *et. al.*, Cancer cell, 2015) in TCGA RNA-seq data (Supplemental Figure 2). For the independent validation data (GSE9891), the lncRNA had similar expression level as that of the protein coding EMT marker genes (Supplemental Figure 5). We explained these two results in the manuscript on pages 5 and 7, respectively. The results show that the selected lncRNA are not expressed at low levels.

4) *MEG3* and *MIAT* are associated with EMT but they are not associated with survival. What is the explanation or speculation from the authors?

Currently, there is no requirement for all the EMT-linked genes to have significant association with overall survival. In fact, only a small proportion of EMT-linked genes show consistent association with overall patient survival in multiple independent cohorts. However, we evaluated the association of five known EMT markers with patient overall survival in three independent ovarian cancer patient cohorts to test this idea. The data show that *SLUG* and *TWIST1* are associated with overall survival, whereas *E-Cadherin*, *N-Cadherin*, and *SNAIL* are not. We have included these data as Supplementary Figure 7. Therefore, having consistently significant association of *DNM3OS* with patient overall survival suggests it is likely to be more clinically important than other EMT-linked markers that are not associated with survival.

What is the correlation between MEG3 and DNM3OS?

As requested, we have evaluated the correlation between *MEG3* and *DNM3OS* in two data sets. In TCGA and the independent validation data GSE9891, Spearman's rank correlation coefficients between *MEG3* and *DNM3OS* are 0.673 and 0.508, respectively. This positive association result was highly likely, as we observed that both lncRNA had elevated expression in mesenchymal subtype compared with the epithelial subtype. However, we are unclear on why the reviewer is asking this question, as these lncRNA are likely to be regulated differently and the expression of one is unlikely to be dependent on the other. Additionally, just because the expression of two genes are associated and one is correlated with survival does not necessarily mean the other has to also be correlated with survival.

5) Did the authors try the linear expression association with survival time for the three lncRNAs (not by median based binary)?

The survival data we present is typically how these data are presented. However, as recommended by the reviewer, we performed linear expression association with survival time for the three lncRNA as a second approach. The results were similar to median-based survival

analysis of the same data sets. Specifically, *DNM3OS*, but not *MEG3* and *MIAT*, had consistently significant negative association with the survival time. These data are now Supplementary Table 5.

6) Also, in the survival association analysis, only univariate was performed. The authors should conduct multivariable analysis with inclusion of other variables (for example stage, treatment, subtype and tumor grade).

Details of tumors were in Table 1 in the first submission of the manuscript and we have left Table 1 in the resubmitted manuscript. Unfortunately, as Table 1 highlights, we were unable to conduct multivariate analysis due to the nature of the data for this study. Specifically, the vast majority of the patient samples in the study were high-grade and late stage serous ovarian cancer. The three patient cohorts (GSE9891, GSE18520, GSE26193) that showed significant association of *DNM3OS* with overall survival have only high-grade tumor samples. For GSE9891, GSE18520, and GSE26193 92%, 100%, and 74% samples are late stage, making the first two data sets not suitable for multivariate analysis after adjusting tumor stage. For GSE26193, *DNM3OS* expression showed a trend associated with poor overall survival; however, the *P*-value is slightly higher than 0.05 significance level (see Fig. 4), and thus, we did not perform multivariate analysis. Additionally, for all the data sets, patient specific treatment information was not available. Finally, for the 700+ patient samples all but 25 were of serous ovarian carcinoma histological subtype, making the numbers too small to properly compare to other ovarian cancer subtypes.

7) What are protein coding genes significantly associated with *DNM3OS* expression? What is their physical relationship between?

In the revised version of the manuscript, we provide a *DNM3OS*-associated protein coding gene list along with the association *P*-values in Supplementary Table 6. We also evaluated physical relationship between the *DNM3OS* associated genes based on the curated protein interaction data available in PINA v2.0 database (Supplementary Table 7). The *DNM3OS* associated protein interaction networks enriched with the EMT linked Gene Ontology biological process terms and the EMT linked canonical signaling pathways (Supplementary Table 8 and 9).

Furthermore, we evaluated predicted binding affinity of *DNM3OS* with the EMT-linked mRNA, which showed significantly stronger binding affinity (Minimum energy score) with *DNM3OS* compared with the genome-wide *DNM3OS*-RNA interaction energy scores. These data are now Figure 5A and Supplementary Figure 8) and described on page 9 of the results in the revised manuscript.

Any adjacent target?

It is not entirely clear to us what this question is asking. We assume the question refers to genes around the *DNM3OS* locus, as some lncRNA have been reported to regulate genes adjacent to where the lncRNA is encoded. *DNM3OS* (*DNM3* opposite strand) is located on the opposite strand of the Dynamin 3 (*DNM3*) gene. The next closest genes 5' and 3' to the *DNM3OS* locus are *METTL13*, *VAMP4*, and *MYOC* that are located 324.6 kb, 406.1 kb, and 470.8 kb away,

respectively. *METTL13* has been reported to be down-regulated in bladder carcinoma and associated with cell migration and invasion (Zhang *et al. Scientific Rep* 2016). However, *METTL13* was not significantly dysregulated in the mesenchymal subtype compared with the epithelial subtype (0.274 and 0.118 fold down; log₂-scale) in TCGA and GSE9891 data, respectively, and was consequently, filtered out from our model system. Currently, there is no evidence for the involvement of the other three genes with ovarian cancer EMT, and they were also excluded systematically from our model system.

8) *If DNM3OS is the target of TWIST1 and it is also coordinated with miR-214, the authors should provide their correlation information.*

We evaluated Spearman's rank correlation and observed strong positive associations between *TWIST1-DNM3OS*, *TWIST1-miRNA*, and *DNM3OS-miRNA*. These data are now Fig. 7C and Fig. 7E in the revised manuscript. Additionally, we also provide experimental results showing that knockdown of *TWIST1* in ovarian cancer cells leads to significantly decreased *DNM3OS* levels (Figure 7B), further supporting the conclusion that *TWIST1* regulates *DNM3OS* expression.

We again thank the reviewers for their critiques. With the added analyses and experimental data, our manuscript is significantly improved from the first submission. Our large scale bioinformatic results and experimental validation continue to show lncRNA that are linked to EMT in ovarian cancer and that *DNM3OS* is overexpressed, correlates to reduced patient survival, and regulates EMT in ovarian cancer. We believe that our important results should now be acceptable for *Nature Communications*.

Sincerely,

Christine M. Eischen, Ph.D.

Professor and Vice Chair, Department of Cancer Biology, Thomas Jefferson University
Co-Leader, Molecular Biology & Genetics Program, Sidney Kimmel Cancer Center

REVIEWERS' COMMENTS:

Reviewer #1 (Remarks to the Author):

The authors have addressed most of my concerns.

Reviewer #2 (Remarks to the Author):

The authors adequately addressed the concerns/comments to the first version. Very appreciate the efforts.